# Appropriate use of blood cultures in the emergency department through machine learning (ABC): study protocol for a randomised controlled non-inferiority trial

Anuschka Y van der Zaag [1], Sheena C Bhagirath,[1] Anneroos W Boerman,[1,2] Michiel Schinkel,[1,3] Ketan Paranjape,[1] Kaoutar Azijli,[4] Milan L Ridderikhof,[5] Mei Lie,[6] Birgit Lissenberg-Witte,[7] Rogier Schade,[8] Joost Wiersinga,[9,10] Robert de Jonge,[2] Prabath W B Nanayakkara[11]

**To cite:** van der Zaag AY, Bhagirath SC, Boerman AW, et al. Appropriate use of blood cultures in the emergency department through machine learning (ABC): study protocol for a randomised controlled non-inferiority trial. BMJ Open 2024;**14**:e084053. doi:10.1136/bmjopen-2024-084053

AYvdZ and SCB contributed equally.

For numbered affiliations see end of article.

**Correspondence to**
Dr Prabath W B Nanayakkara;
p.nanayakkara@amsterdamumc.nl

## ABSTRACT

**Introduction** The liberal use of blood cultures in emergency departments (EDs) leads to low yields and high numbers of false-positive results. False-positive, contaminated cultures are associated with prolonged hospital stays, increased antibiotic usage and even higher hospital mortality rates. This trial aims to investigate whether a recently developed and validated machine learning model for predicting blood culture outcomes can safely and effectively guide clinicians in withholding unnecessary blood culture analysis.

**Methods and analysis** A randomised controlled, non-inferiority trial comparing current practice with a machine learning-guided approach. The primary objective is to determine whether the machine learning based approach is non-inferior to standard practice based on 30-day mortality. Secondary outcomes include hospital length-of-stay and hospital admission rates. Other outcomes include model performance and antibiotic usage. Participants will be recruited in the EDs of multiple hospitals in the Netherlands. A total of 7584 participants will be included.

**Ethics and dissemination** Possible participants will receive verbal information and a paper information brochure regarding the trial. They will be given at least 1 hour consideration time before providing informed consent. Research results will be published in peer-reviewed journals. This study has been approved by the Amsterdam University Medical Centers' local medical ethics review committee (No 22.0567). The study will be conducted in concordance with the principles of the Declaration of Helsinki and in accordance with the Medical Research Involving Human Subjects Act, General Data Privacy Regulation and Medical Device Regulation.

**Trial registration number** NCT06163781.

## INTRODUCTION

Over 20% of adult emergency department (ED) visits occur due to infections.[1] Often, physicians will request blood cultures in these patients due to fear of a bloodstream

---

## STRENGTHS AND LIMITATIONS OF THIS STUDY

⇒ Pioneering the use of machine learning in a randomised controlled trial to predict blood culture outcomes.
⇒ Conducting a comparative analysis against current clinical practices.
⇒ Assessing patient-centred outcomes and performing an economic evaluation.
⇒ May restrict future applications due to the exclusion of immunocompromised patients.

---

infection (BSI).[2] Because of the liberal use of blood cultures, the yield tends to be low. Depending on the setting, only about 1%–15% of blood cultures grow pathogens, of which 30%–55% turn out to be false-positive, contaminated results.[3–5] Those contaminated cultures are associated with prolonged hospital stays, unnecessary use of antibiotics and even hospital mortality.[3 6 7] To optimise blood culture use, it is crucial to better target which patients would benefit from blood culture analysis.[8]

A machine learning model based on extreme gradient-boosting (XG Boost) that can predict the outcome of blood cultures in the ED has been developed to address this clinical dilemma, aiming to reduce unnecessary tests in low-risk patients, and help avoid the harms associated with false-positive results due to blood culture contamination.[9 10] The performance of the model has been internally and externally validated in various settings, including a real-time background evaluation in the Amsterdam University Medical Centers (Amsterdam UMC, locations VUmc and AMC) Electronic health record (EHR)

system, where it maintained an area under the operating curve of 0.76 and showed a potential reduction of approximately 30% blood cultures in the future.[9 10] Additionally, the model has shown a consistent performance over time despite changes in clinical practice and patient characteristics.[11]

The aim of this study is to investigate the safety and potential clinical benefits of using the blood culture prediction tool as decision support in reducing unnecessary blood cultures and thereby the negative side effects of false-positive results for patients. This will be the first randomised controlled trial testing such a machine learning prediction tool in clinical practice.

## METHODS AND ANALYSIS

This study protocol adheres to the Standard Protocol Items: Recommendations for Interventional Trials statement.

### Model development and validation

This segment features a summary of the model development, more detailed descriptions have been published previously.[9 10] For the development of the model data from the VUmc was split into a training (80%) and a test set (20%), stratified by blood culture outcomes. Missing data were imputed by medians. The model was trained on the training set and subsequently validated in the VUmc test set, and three other hospital datasets (AmsterdamUMC location AMC, Zaans Medisch Centrum (ZMC, Zaandam, The Netherlands) and Beth Israel Deaconess Medical Center (BIDMC, Boston, Massachusetts, USA)).

These hospitals differ in size and population, comprising two tertiary teaching hospitals and one secondary care hospital in the Netherlands, along with one hospital dataset from the USA. Part of the data was gathered during the COVID-19 pandemic, but this did not affect its performance. The model contains 49 variables, namely age, sex, 6 vital sign measurements and 18 laboratory tests. Additionally, indicator variables of the laboratory tests were included. Figure 1 shows the 20 most important variables. The area under the receiver operating characteristic (AUROC) curves in the various validation cohorts ranged between 0.75 and 0.81. A prospective validation was also performed, integrating the model into the EHR of VUmc, showing an AUROC of 0.76. With the threshold for blood culture analysis set at 5%, analyses could be avoided at approximately 30%.

### Study design

This study will be a multicentre randomised controlled non-inferiority trial. Participants will be randomised into an intervention and control arm. In the intervention group, the blood culture prediction model is used to predict the probability of a positive blood culture in the patient. The model will start predicting when sufficient variables are available in the EHR. A new prediction will be made every 20 min until 3 hours have passed since check—in of the patient in the ED. The predictions made by the model will be shown to the physician after randomisation in a dashboard integrated into the EHRs. The blood culture analysis will be cancelled if the probability of a positive culture is less than 5%. This will be

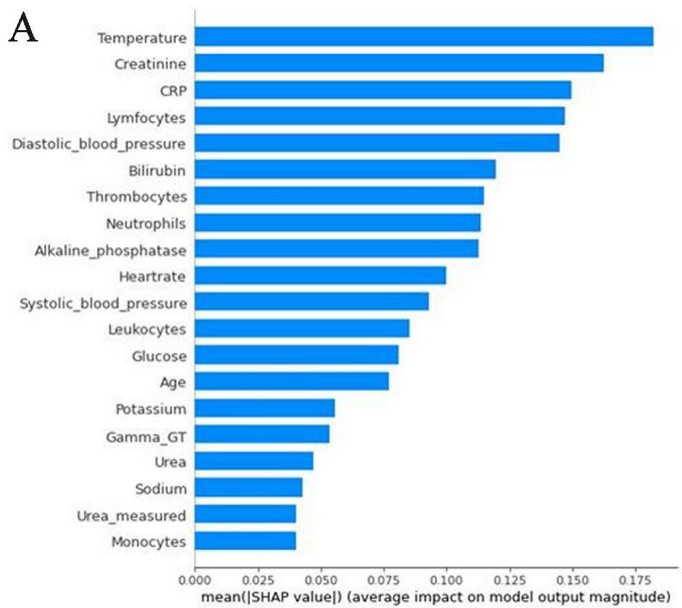
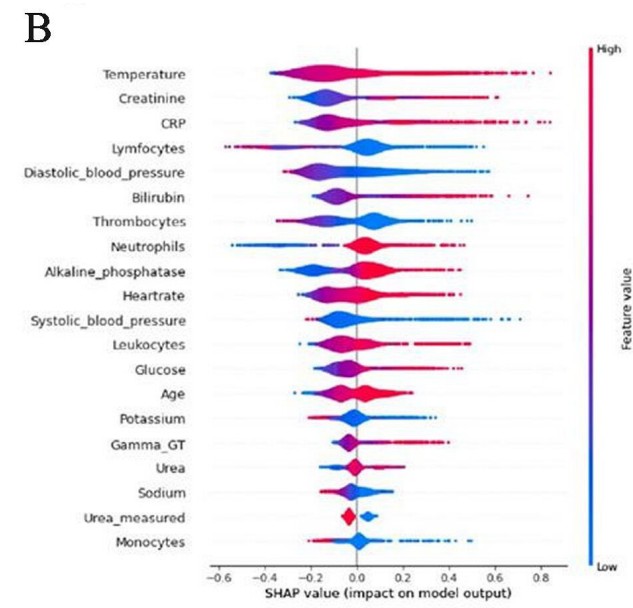

**Figure 1** SHAP values for model variables. This figure depicts SHAP (SHapley Additive exPlanation) values, illustrating the individual contributions of each variable to the model's prediction. In the left panel (blue bars), we see the average impact of the 20 most important features on the prediction. Meanwhile, the right panel not only shows the feature's contribution but also its direction. Contributions to the left of 0 on the x-axis correspond to negative predictions for blood cultures, while those to the right correspond with positive predictions. Additionally, the colour indicates the predictor's actual value, with blue indicating low values and red indicating high values.

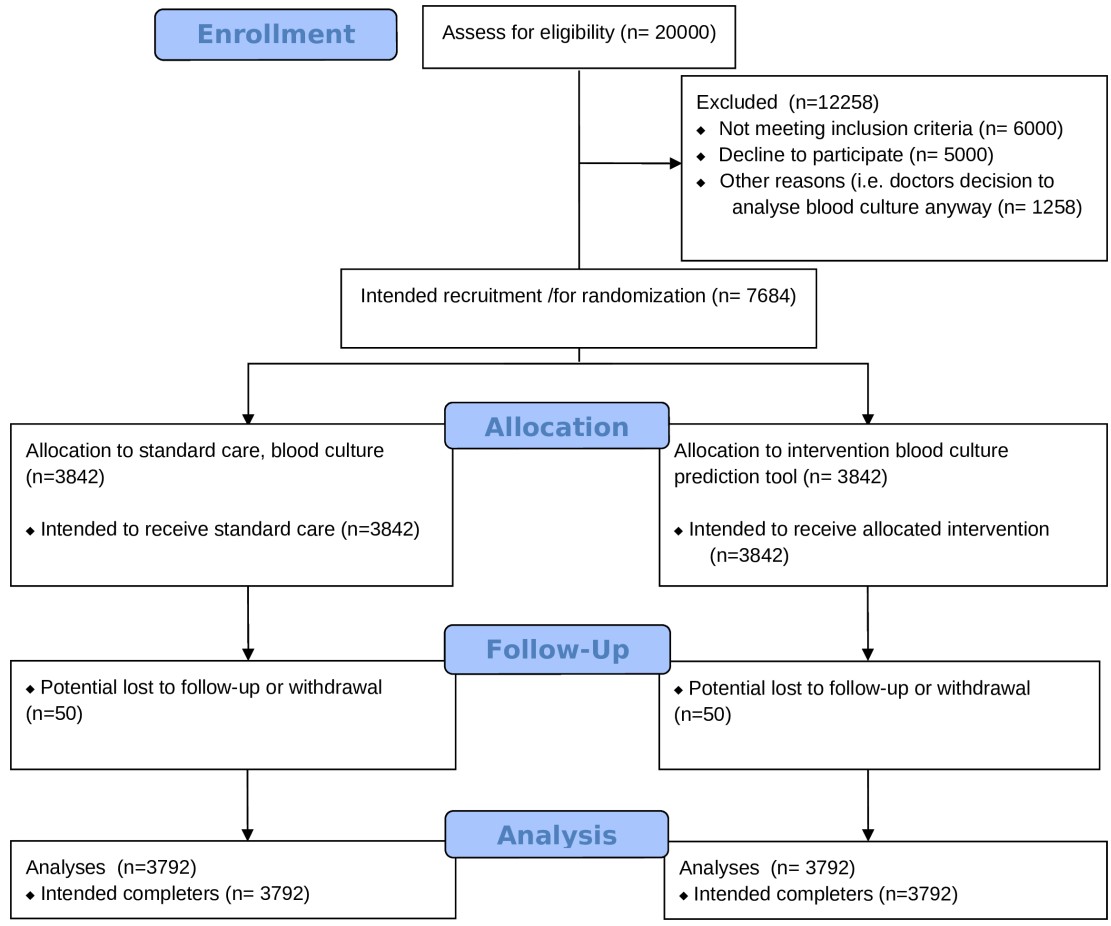

**Figure 2** Consolidated Standards of Reporting Trials flow chart.

done manually by the study team. If the probability is over or equal to 5%, the blood culture will be analysed as per standard practice. In the control group, all patients will have their blood cultures analysed. The study design is summarised in figures 2 and 3. Patients will be randomised within the EHR system, to ensure that physicians cannot be influenced in their treatment decisions by seeing the score before the patient is included. The first patient was included in February 2024. Follow-up of the last patients is expected in February 2027.

## Population

All adult patients (aged 18 years or older) presenting to the EDs of multiple hospitals in the Netherlands, and in whom blood cultures are ordered during ED stay will be assessed for eligibility. This process is automated within the EHR. The treating physician determines whether there is an indication for performing blood cultures according to the current local standards, after which both the patient and study team are informed about possible study recruitment. Inclusion and exclusion criteria are summarised in table 1. Some patients will be excluded due to the higher possibility of BSIs with a pathogen generally considered a contaminant (eg, a central line in situ) or because of the possibility of severe clinical implications of omitting blood cultures (eg, severe neutropenia). Since the study's goal is to reduce unnecessary testing, patients for whom

blood culture analysis is deemed imperative are excluded (eg, patients with a suspected diagnosis of endocarditis, spondylodiscitis or infected prosthetic material). Furthermore, patients unable to give informed consent and pregnant or breastfeeding patients will be excluded, due to them being considered high-risk groups by the ethics review board. Therefore, the review board did not give permission to include these groups.

## Sample size calculation

An analysis of 5907 unique ED visits with blood culture sampling in Amsterdam UMC showed a 30-day mortality rate of 7.6%, a hospital admission rate of 67.4% and an average length of stay in the hospital of 6.7 days. Based on these numbers, the sample sizes with a relative non-inferiority margin of 1.25 for the rates and 1 day for the length of stay were calculated. The sample sizes were calculated by a statistician based on a one-sided alpha of 5% and a target power of at least 80%. For 30-day mortality, the calculated sample size was 6066 (3033 per arm) to test non-inferiority. After inflation for a dropout rate of 20%, the final sample size is 7584 (3792 per arm). For the secondary outcomes, described in table 2, fewer participants were needed for adequate power.

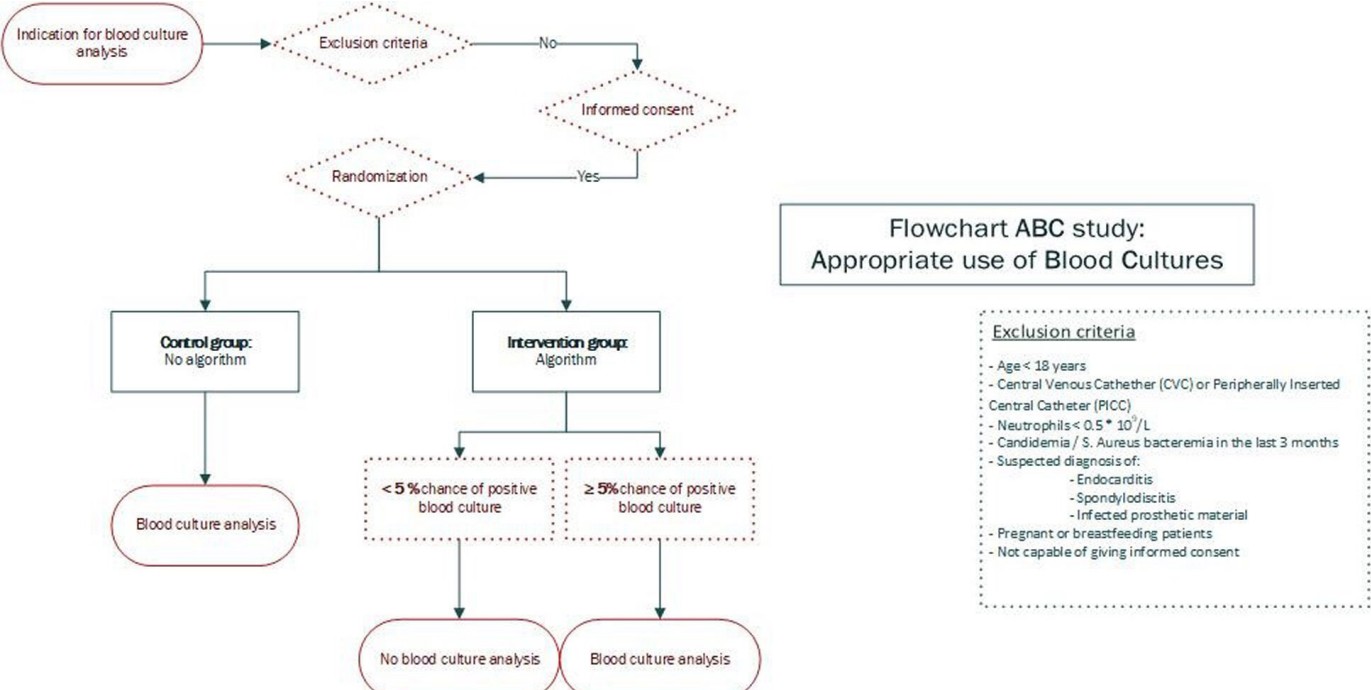

**Figure 3** Study design flow chart.

## Recruitment

In order to achieve adequate participant enrolment, an announcement regarding the study is built into the EHR system. When the physician initiates a blood culture request in the EHR for an adult patient in the ED, a pop-up notification will automatically appear. This pop-up briefly outlines the study and presents both inclusion and exclusion criteria. If the patient provides permission to

receive information about the study, the study team will be alerted and will provide the patient with additional information and obtain informed consent.

## Outcomes

The primary objective is to investigate whether using a machine learning-based blood culture prediction tool is non-inferior to current practice regarding 30-day all-cause mortality. All secondary outcomes are described in table 2. Model performance will be evaluated using the AUROC, Area Under the Precision-Recall Curve (AUPRC) and calibration for the model in the complete study population and subgroups based on comorbidities. This will be assessed every 3 months so model performance over time can be guaranteed. The model will also continue to undergo prospective validation in the background of the EHR. Additionally, patient-reported outcomes will be included to determine if the model also influences patient-reported outcome measures such as side effects from antibiotics and quality of life. For assessment of quality of life, the EuroQol 5-Dimension 5-level (EQ-5D-5L) Questionnaire[12] will be filled out at baseline and 3 months. Lastly, a cost-effectiveness analysis will be performed using cost questionnaires based on the iMTA Medical Consumption Questionnaire (iMCQ) and iMTA Productivity Cost Questionnaire (iPCQ).[13] A timeline visualising data collection is shown in table 3.

## Data collection and management

The data underlying this study, including the outcome measures such as mortality and admission rates, will be automatically extracted from the EHR system into an electronic data capture system (Castor EDC). The questionnaires regarding patient-reported outcomes and

| Table 1 | Overview of inclusion and exclusion criteria |
|---|---|
| Inclusion criteria | Age ≥18 years |
| | Willing and able to give informed consent to participate |
| | Have a clinical indication for blood culture analysis (according to the treating physician) |
| | Have sufficient data recorded (laboratory results and vital sign measurements) for a prediction to be made* |
| Exclusion criteria | Patients with a central venous catheter or peripherally inserted central catheter in situ |
| | Neutrophil count <0.5×10⁹/L |
| | Candidemia or *Staphylococcus aureus* bacteremia in the past 3 months |
| | Suspected endocarditis, spondylodiscitis or infected prosthetic material |
| | Pregnant or breastfeeding patients |
| | Inability to give informed consent |

*Predictions can only be made when at least 20% of the needed parameters are available for a patient, which is the case in approximately 80% of all patients who undergo blood culture analyses in the emergency department.

**Table 2** Overview of outcome measures

| | |
|---|---|
| Primary outcome | 30-day all-cause mortality |
| Key secondary outcomes | In-hospital mortality |
| | Hospital admission rates |
| | Hospital length-of-stay |
| Other outcome measures | |
| Patient-related outcomes | 90-day mortality |
| | 30-day readmission rates |
| | Emergency department length-of-stay in hours |
| Diagnostics related outcomes | Percentage of blood cultures avoided in the intervention group |
| | Number of blood cultures on each day of hospital stay (in admitted patients) |
| | Percentage of positive blood cultures in each group |
| | Total number of laboratory and microbiology tests in the ED |
| | Total number of laboratory and microbiology tests on each day of hospital stay (in admitted patients) |
| Therapy-related outcomes | Percentage of patients receiving antibiotics in the ED |
| | Duration of antibiotic therapy |
| | Types of antibiotics given in the ED |
| Model-related outcomes | Model performance (AUC) during the trial |
| | Model performance in subgroups:<br>▶ Immunocompromised patients (triple immunosuppressive therapy)<br>▶ Patients who had transplants |

AUC, area under the operating curve; ED, emergency department.

cost–benefit analysis will subsequently be sent out digitally through the EDC system. The data will be stored for 15 years, in agreement with local regulations and with consent. Data management and monitoring will be performed by the study team. Additionally, a monitor from the local independent clinical monitoring centre will monitor the data and perform data validation checks.

Participant data underlying the results of this study can be shared. The data can be requested following publication of this work. The data can be shared with researchers on reasonable request, which is allowed under local privacy regulations. Proposals should be directed to the corresponding author and requestors will need to sign a data access agreement.

**Table 3** Data collection timeline

| Allocation | Enrolment | Allocation | Baseline | During admission* | 3 months after baseline |
|---|---|---|---|---|---|
| Eligibility screen | X | | | | |
| Informed consent | X | | | | |
| Allocation | | X | | | |
| Vital parameters | | | X | | |
| Lab values | | | X | | |
| Quality of life (according to EQ-5D-5L) | | | X | | X |
| Cancelling blood culture[A] | | | X[A] | | |
| Admission rates | | | X | | |
| Mortality | | | | X | X |
| Patient-related outcomes[B] | | | X | X | X |
| Diagnostics related outcomes | | | X | X | X |
| Therapy-related outcomes | | | X | X | X |
| Model related outcomes | | | X | X | X |
| Cost questionnaire | | | | | X |

A: Only in intervention group when predicted probability of positive blood culture is <5%. B: See table 2.
*Not applicable to all participants.

## Safety, adverse events and monitoring

Adverse events will be monitored by the investigators, including device-related adverse events. Serious adverse events will be reported within a maximum of 15 days to the local authority, and an annual safety report will be submitted to the local authorities. Examples of possible adverse events include delayed appropriate antibiotic treatment, hospital readmission or prolonging of hospital stay. The study risk was assessed using the Nederlandse Federatie van Universitair Medische Centra (NFU) risk classification and classified as moderate risk.[14] The study has its own data safety monitoring board (DSMB) consisting of clinicians, a statistician and an epidemiologist, independent of the study team. The DSMB will meet before the first year of the study ends and once a year during the study period. During the study period, the DSMB will be supplied with an interim analysis and will advise the study team based on these results. A meeting will be held to evaluate mortality after 2850 patients have completed the trial. Additional meetings may be requested based on trial events. At least 2 weeks before each meeting, the DSMB will receive reports regarding data quality, recruitment and patients' safety. The DSMB will report back to the principal investigator regarding any advice they may have. Additionally, the study will be monitored by an independent clinical trial monitoring committee, which performs quality checks on the data and regular on-site auditing.

## Allocation and randomisation

Participants will be randomly assigned to either the intervention or control group with a 1:1 allocation through a validated computer-generated random number within the EHR system. This will be a simple randomisation without stratification. Randomisation will occur automatically after consent and after blood cultures have been ordered. Physicians caring for a patient in the control group will not see any algorithm predictions. Physicians caring for a patient in the intervention group will be able to see the algorithm prediction. Due to the nature of the intervention, neither participant nor staff can be blinded to the allocation.

## Statistical analysis

We will carry out a per-protocol analysis. The primary outcome, the 30-day mortality rate, will be compared between the groups using a non-inferiority test for the ratio of two proportions. This will be a one-sided test with a significance level of 0.05. Additionally, we will do a multivariable analysis of the primary and key secondary outcomes to adjust for potential effect of age, Charlson Comorbidity Index, use of immunosuppressive medication, Modified Early Warning Score and resuscitation status. Similar tests will be performed for in-hospital mortality rate and the hospital admission rate. The hospital length-of-stay will be analysed as a continuous variable and a non-inferiority test for the differences between the two means will be used. The test statistic to be used is a one-sided T-test with a significance level of 0.05. The model-related outcomes will be assessed using an area under the curve and calibration plots. Baseline characteristics will be presented using descriptive statistics, without statistical testing for differences, since this is a randomised trial.

## Patient and public involvement

Patient participation is important in both the clinical trial and the implementation of Artificial Intelligence (AI) in clinical practice. The Client Advisory Board of Amsterdam UMC and a former sepsis patient evaluated the study question as relevant and the study design as achievable from patients' perspective. They will stay involved during the study as well. Also, to disseminate results of the study, we will collaborate with relevant patient organisations. Furthermore, a patient safety and implementation expert was involved in the development of the study protocol and will remain involved during the study period. They will continue to stay involved if the study is deemed successful and the model can be implemented into clinical practice.

## ETHICS AND DISSEMINATION

Obtaining informed consent is of the utmost importance when working with study participants. In order to ensure that participants are well informed about the study before consenting, all potential participants will have the trial clearly explained to them by a trained member of the study team. In addition to the verbal information, a paper information brochure will be provided to each participant, see online supplemental material 1. We inform patients about the procedures and what the consequences of doing or withholding blood cultures can be, as this is the most relevant information for them. Each participant will be made aware of their right to withdraw consent at any given time. If a patient withdraws consent their data will no longer be used in the study and the study personnel will not follow them up. If the patient was in the intervention group and blood culture analysis was already cancelled, we cannot reinstate the analysis, since the material has been destroyed. Due to the nature of the study, the potential participant will get 1 hour to consider joining the study. If any change to the protocol is made the ethics review board will be notified and will have to give permission. If the changes are relevant to the participants, they will receive additional information regarding the changes made.

Research results will be published in peer-reviewed journals approximately 1 year after the trial has ended. This will give the research team sufficient time for statistical analyses.

This study will be conducted according to the principles of the Declaration of Helsinki and in accordance with the Medical Research Involving Human Subjects Act, General Data Privacy Regulation and Medical Device Regulation. The study was approved by the Amsterdam University Medical Center's medical ethics review committee with number 22.0567. The trial was registered at Clinicaltrials. gov (NCT06163781).

## DISCUSSION

Blood cultures are a commonly used diagnostic tool in the ED. Unfortunately, low yields and high rates of contamination lead to unnecessary use of antibiotics, prolonged length-of-stay in the hospital and even higher mortality. A machine learning model has been validated retrospectively and prospectively and can identify patients at low risk for positive blood cultures. The study described in this protocol is the first randomised controlled trial using machine learning to predict blood culture outcomes. However, possible limits to this trial include restricted future application due to the exclusion of severely immunocompromised patients. Future studies may need to focus on this specific subgroup as well. Nevertheless, this study will include a heterogeneous group presenting to the ED and may greatly influence current standards of practice. By using this model in practice, it can potentially reduce a significant amount of blood culture analyses and prevent the undesirable effects of false-positive blood culture results.

### Author affiliations
[1]Department of Internal Medicine, Division of Acute Medicine, Amsterdam UMC Locatie VUmc, Amsterdam, The Netherlands
[2]Department of Laboratory Medicine, Amsterdam UMC Locatie VUmc, Amsterdam, The Netherlands
[3]Center for Experimental and Molecular Medicine (C.E.M.M.), Amsterdam University Medical Centres, Amsterdam, The Netherlands
[4]Department of Emergency Medicine, Amsterdam UMC Locatie VUmc, Amsterdam, The Netherlands
[5]Department of Emergency Medicine, Amsterdam UMC Locatie AMC, Amsterdam, The Netherlands
[6]Department of EVA Service Center, Amsterdam UMC Locatie VUmc, Amsterdam, The Netherlands
[7]Department of Epidemiology & Data Science, Amsterdam UMC Locatie VUmc, Amsterdam, The Netherlands
[8]Department of Medical Microbiology and Infection Prevention, Amsterdam UMC Locatie VUmc, Amsterdam, The Netherlands
[9]Department of Internal Medicine, Division of Infectious Diseases, Amsterdam UMC Locatie AMC, Amsterdam, The Netherlands
[10]Center for Experimental and Molecular Medicine (C.E.M.M.), Amsterdam UMC Locatie AMC, Amsterdam, The Netherlands
[11]Department of Internal Medicine, Section General Internal Medicine, Amsterdam UMC Locatie VUmc, Amsterdam, The Netherlands

**Contributors** AYvdZ and SCB contributed equally to this paper. AYvdZ and SB wrote the first draft of the paper. AWB, MS and PWBN conceptualised the study. AWB, MS, JW and PWBN secured funding for the study. BL-W provided the sample size calculations and statistical analysis plan. ML developed the resources. KP, RS, KA, MLR, ML, JW, RdJ and PWBN supervised various parts of the research process within their expertise. AYvdZ, SCB, AWB, MS, MLR and PWBN revised the manuscript. All authors approved the final version of the manuscript. The prediction model is an AI tool, specifically machine learning. For the writing of the manuscript, AI was only used for grammar-checking of a few sentences.

**Funding** This RCT is funded by ZonMW DoelmatigheidsOnderzoek, grant number 2012379. Additionally, this trial is supported by a TKI grant, which aims to use these trial results to further optimize the tool. The funders had no role in the study design and will have no role in its execution, data analyses or decision to publish.

**Competing interests** None declared.

**Patient and public involvement** Patients and/or the public were involved in the design, or conduct, or reporting, or dissemination plans of this research. Refer to the Methods section for further details.

**Patient consent for publication** Not applicable.

**Provenance and peer review** Not commissioned; externally peer reviewed.

**ORCID iD**
Anuschka Y van der Zaag http://orcid.org/0009-0003-4358-3242

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
