## [Reviewer comments · BMJ Open]

ARTICLE DETAILS

TITLE (PROVISIONAL)	Appropriate use of Blood Cultures in the emergency department through machine learning (ABC): study protocol for a randomized controlled non-inferiority trial
AUTHORS	van der Zaag, Anuschka; Bhagirath, Sheena C.; Boerman, Anneroo; Schinkel, Michiel; Paranjape, Ketan; Azijli, Kaoutar; Ridderikhof, Milan L.; Lie, Mei; Lissenberg-Witte, Birgit; Schade, Rogier; Wiersinga, Joost; de Jonge, Robert; Nanayakkara, Prabath

VERSION 1 – REVIEW

REVIEWER	McFadden, Benjamin R. The University of Western Australia Faculty of Engineering Computing and Mathematics
REVIEW RETURNED	25-Jan-2024

GENERAL COMMENTS	Firstly, I would like to applaud the authors on presenting an ambitious, well written and important protocol. The area of blood culture outcome prediction using machine learning is a growing area of research, and this proposed study would contribute significantly to the literature, with potential outcomes for implementation in practice. I have identified some areas for minor revision, which I believe would further improve the clarity of the protocol. ** Revisions ** - It would be useful to include the machine learning model that is being used in the study. From reading the authors previous work, it appears that the model in question is an XGBoost model. If this is still the case, it would be helpful for readers to know this from reading the protocol. I would include mention of the model that you are using in the introduction where you have referenced the previous work.- Similarly, including a table with the parameters / features required for the machine learning model would improve the clarity, and support the potential reproduction of the protocol.- Lastly, the exclusion criteria in Box 1 mentions the exclusion of "Pregnant or breastfeeding patients". However, this criteria is not included in exclusion criteria box in figure 1. This criteria is also not mentioned on the clinicaltrials.gov registration. It would be useful if the authors could clarify this component of the protocol.
---

REVIEWER	Vijlbrief, Daniel (D.C.)
-----------------	--------------------------

	Wilhelmina Children's Hospital
REVIEW RETURNED	11-Feb-2024

GENERAL COMMENTS	With great interest, I have reviewed the manuscript submitted by van der Zaag et al., which outlines a study protocol for a randomized controlled non-inferiority trial. The trial aims to assess the effectiveness and safety of a machine learning model designed to guide the use of blood cultures in emergency departments (EDs). The primary goal is to reduce unnecessary blood culture analyses by predicting blood culture outcomes, potentially decreasing false-positive rates, antibiotic misuse, prolonged hospital stays, and even mortality rates associated with contaminated cultures. The aim of the study is highly commendable and I wish the researchers a lot of fortune in investigating this procedure. Major Comments:  1. Validation and Adaptability of the Machine Learning Model: The manuscript briefly mentions the validation of the machine learning model but lacks detail on how the model was trained, the features being used, and how it might adapt to different hospital settings or populations. Given the variability in clinical practices and patient demographics across hospitals, elaborating on these aspects would significantly strengthen the protocol. Have different procedures in the different hospitals been analyzed, are protocols relatively the same? Any difference in EHRs between hospitals. 2. Management of Model Drift: Machine learning models may experience performance degradation over time due to changes in clinical practices, fi over the course of 4 years, pathogen prevalence, or patient demographics. The manuscript could discuss strategies for monitoring and updating the model to maintain its accuracy and relevance. 3. Inclusion and Exclusion Criteria: While the manuscript outlines the criteria, it may benefit from a more detailed discussion on the rationale behind specific exclusions, such as patients with certain comorbidities. Clarifying these choices would help in understanding the study population's representativeness. 4. Ethical Considerations: The use of machine learning in clinical decision-making raises ethical questions, especially regarding transparency, accountability, and patient consent. The manuscript should elaborate on how patients are informed about the AI component in their care and any measures to address potential ethical concerns. The process seems quite automated, but if the patients wants to withdraw consent, what happens then? 5. Safety What exactly will be monitored to address safety, what are the SAEs, are there any unexpected adverse events to be expected? At what moments will the DSMB come together? Minor Comments:  • Technical Specifications: Additional details about the machine learning model, including the algorithm used, features, PPV/NPV/Sens/Spec etc are interesting for readers of the protocol.
---

	 • Statistical Analysis: More information on the statistical methods for comparing primary and secondary outcomes between the intervention and control groups would enhance the manuscript's methodological clarity. • Patient and Public Involvement: The manuscript mentions involving a former sepsis patient in evaluating the study question. Expanding on how patient feedback influenced the study design could highlight the value of patient-centered research approaches. Conclusion: The study protocol presented by van der Zaag et al. addresses an important clinical challenge in emergency medicine. By integrating machine learning into the decision-making process for blood culture analysis, this research has the potential to optimize patient care and resource utilization in EDs. However, addressing the major and minor comments outlined above could enhance the manuscript's strength, clarity, and impact.
--	---

VERSION 1 – AUTHOR RESPONSE

Reviewer 1

Comment 1:

It would be useful to include the machine learning model that is being used in the study. From reading the authors previous work, it appears that the model in question is an XGBoost model. If this is still the case, it would be helpful for readers to know this from reading the protocol. I would include mention of the model that you are using in the introduction where you have referenced the previous work.

Author response 1:

We agree that details regarding model development and specifics are helpful for readers and were lacking in the first version of the manuscript. They have been added in the section 'Model development and validation' on page 4, starting at line 83.

Comment 2:

Similarly, including a table with the parameters / features required for the machine learning model would improve the clarity, and support the potential reproduction of the protocol.

Author response 2:

A table with the 20 most important model parameters was added (Figure 1)

Comment 3:

Lastly, the exclusion criteria in Box 1 mentions the exclusion of "Pregnant or breastfeeding patients". However, this criteria is not included in exclusion criteria box in figure 1. This criteria is also not mentioned on the clinicaltrials.gov registration. It would be useful if the authors could clarify this component of the protocol.

Author response 3:

We have included this exclusion criteria in Box 1 and have added a short clarification in the Methods section, in the population subsection on page 5, starting from line 115. The criteria were also added to Clinicaltrials.gov, this is awaiting approval.

Reviewer 2

Major Comment 1:

Validation and Adaptability of the Machine Learning Model: The manuscript briefly mentions the validation of the machine learning model but lacks detail on how the model was trained, the features being used, and how it might adapt to different hospital settings or populations. Given the variability in clinical practices and patient demographics across hospitals, elaborating on these aspects would significantly strengthen the protocol. Have different procedures in the different hospitals been analyzed, are protocols relatively the same? Any difference in EHRs between hospitals.

Author response major comment 1:

We acknowledge that the information provided regarding model training and features was too limited and have provided additional information in the section 'Model development and validation', page 4 lines 83-96. Also, additional information regarding validation in varying clinical settings was added. We would like to ask the reviewer whether this information is sufficient or additional clarifications are needed?

Major Comment 2:

Management of Model Drift: Machine learning models may experience performance degradation over time due to changes in clinical practices, fi over the course of 4 years, pathogen prevalence, or patient demographics. The manuscript could discuss strategies for monitoring and updating the model to maintain its accuracy and relevance.

Author response major comment 2:

We agree that insufficient information was given in the manuscript regarding management of model drift. We have added information regarding model performance assessment in the Methods section, page 6, lines 140-143.

Major Comment 3:

Inclusion and Exclusion Criteria: While the manuscript outlines the criteria, it may benefit from a more detailed discussion on the rationale behind specific exclusions, such as patients with certain comorbidities. Clarifying these choices would help in understanding the study population's representativeness.

Author response major comment 3:

We have further clarified the rationale behind specific exclusion criteria in the Methods section on pages 4-5, lines 115-122.

Major Comment 4:

Ethical Considerations: The use of machine learning in clinical decision-making raises ethical questions, especially regarding transparency, accountability, and patient consent. The manuscript should elaborate on how patients are informed about the AI component in their care and any measures to address potential ethical concerns. The process seems quite automated, but if the patients wants to withdraw consent, what happens then?

Author response major comment 4:

In the participant information, we do mention the fact that AI is used, but we do not put much emphasis on the specific techniques, as this is not relevant or understandable for the patients (which was also the opinion of our patient representative). We have added information regarding this on page 9, lines 209-214. We have added information about what happens if a patient wishes to withdraw their consent on page 7, lines 214-217.

Major Comment 5:

What exactly will be monitored to address safety, what are the SAEs, are there any unexpected adverse events to be expected? At what moments will the DSMB come together?

Author response major comment 5:

We have added additional information regarding safety measures taken during the trial on page 7-8.

Minor comment 1:

Technical Specifications: Additional details about the machine learning model, including the algorithm used, features, PPV/NPV/Sens/Spec etc are interesting for readers of the protocol.

Author response minor comment 1:

We have added additional details about the model, including the type of algorithm and features in the section on model development on page 4. We have chosen to not also add PPV/NPV/Sens/Spec since we feel that the AUC and data regarding the prospective validation of the model are of greater significance.

Minor comment 2:

Statistical Analysis: More information on the statistical methods for comparing primary and secondary outcomes between the intervention and control groups would enhance the manuscript's methodological clarity.

Author response minor comment 2:

We have tried to provide additional information regarding the statistical methods and rewritten part of the section in order to make it more clear on page 8. We would like to ask the reviewer if the section is clear now or if additional information is still warranted?

Minor comment 3:

Patient and Public Involvement: The manuscript mentions involving a former sepsis patient in evaluating the study question. Expanding on how patient feedback influenced the study design could highlight the value of patient-centered research approaches.

Author response minor comment 3:

We agree with the reviewer that this was not described sufficiently. However, there were no changes made to the study design based on the feedback we received from the patients. They will remain involved during the study, and if any changes need to be made they will be involved, for example by giving feedback in case a new informed consent form is needed.

VERSION 2 – REVIEW

REVIEWER	McFadden, Benjamin R. The University of Western Australia Faculty of Engineering Computing and Mathematics
REVIEW RETURNED	31-Mar-2024

GENERAL COMMENTS	All of my previous comments have been addressed sufficiently by the authors. As I have no further comments, I recommend that this protocol be accepted for publication. I wish all the authors the best of luck with completing a successful trial.
--

REVIEWER	Vijlbrief, Daniel (D.C.) Wilhelmina Children's Hospital
REVIEW RETURNED	24-Mar-2024

GENERAL COMMENTS	I would like to thanks the authors for there answers to my concerns. The manuscript, especially including the model specifics, improved the manuscript. One small item, the ZMC and BIDMC are not defined, a Dutch reader might directly identify these hospitals.
---

VERSION 2 – AUTHOR RESPONSE

Regarding comments by reviewer 1.

We thank reviewer 1 for their recommendation that this protocol be accepted for publication, and again for previous comments that ameliorated this manuscript.

Regarding comments by reviewer 2.

We thank reviewer 2 for the compliment regarding the manuscript improvement. We have defined the other two hospitals in the manuscript for clarity. There was no intention not to disclose the hospitals.